# Measurement of the transition frequency from $2S_{1/2}, F = 0$ to $2P_{1/2}, F = 1$ states in Muonium

Gianluca Janka [1], Ben Ohayon[1], Irene Cortinovis[1], Zak Burkley[1], Lucas de Sousa Borges [1], Emilio Depero[1], Artem Golovizin [1,2], Xiaojie Ni[3], Zaher Salman [3], Andreas Suter [3], Thomas Prokscha [3] & Paolo Crivelli [1] ✉

Muons are puzzling physicists since their discovery when they were first thought to be the meson predicted by Yukawa to mediate the strong force. The recent result at Fermilab on the muon g-2 anomaly puts the muonic sector once more under the spotlight and calls for further measurements with this particle. Here, we present the results of the measurement of the $2S_{1/2}$, $F = 0 \rightarrow 2P_{1/2}, F = 1$ transition in Muonium. The measured value of 580.6(6.8) MHz is in agreement with the theoretical calculations. A value of the Lamb shift of 1045.5(6.8) MHz is extracted, compatible with previous experiments. We also determine the $2S$ hyperfine splitting in Muonium to be 559.6(7.2) MHz. The measured transition being isolated from the other hyperfine levels holds the promise to provide an improved determination of the Muonium Lamb shift at a level where bound state QED recoil corrections not accessible in hydrogen could be tested. This result would be sensitive to new physics in the muonic sector, e.g., to new bosons which might provide an explanation of the g-2 muon anomaly and allow to test Lorentz and CPT violation. We also present the observation of Muonium in the $n = 3$ excited state opening up the possibility of additional precise microwave measurements.

The discovery of muons has an intriguing history. They were first detected in cosmic radiation by Anderson and Neddermayer in 1936[1]. Interestingly, the first hints of muons were already seen in 1933 by Kunze[2] in his Wilson chamber, but he was not confident enough about his results to claim the discovery of a novel particle.

The muon was first thought to be the pion predicted by Yukawa[3], the heavy quantum (meson) responsible of mediating the nuclear (strong) force in analogy to the light quantum (photon) for the electromagnetic interaction. The pion, based on the range of the nuclear force, should have had a mass of 100–200 times the mass of the electron and should be both positively and negatively charged. The muon (207 times the electron mass) seemed to have just the expected value. However, in 1946 an experiment of Conversi et al.[4] showed that their interaction with nuclei was too weak to be attributed to pions.

They observed that the negative mesotron would decay instead of being absorbed by carbon after having formed a pionic atom as predicted by Tomonaga-Araki[5]. This was the first indication of the formation of muonic atoms as it was realized a few years later. Moreover, the lifetime of the muon was of the order of $10^{-6}$ s which is $10^{12}$ times longer than expected by strong interaction processes. Finally in 1947, Powell et al.[6] detected the pion using photographic emulsions at high altitudes and verified that it decays into a muon and a neutral particle that was identified to be a neutrino. This led Isidor Rabi to come up with his famous quote: "Who ordered that?".

The fascinating history of the muon continues to this day. The recent results at Fermilab[7] confirming the results at Brookhaven National Laboratory[8] that the measured muon anomalous magnetic moment (g-2 muon) deviates from the standard model prediction[9] by

[1]Institute for Particle Physics and Astrophysics, ETH Zürich, CH-8093 Zürich, Switzerland. [2]P.N. Lebedev Physical Institute, 53 Leninsky prospekt., Moscow 119991, Russia. [3]Laboratory for Muon Spin Spectroscopy, Paul Scherrer Institute, CH-5232 Villigen PSI, Switzerland. ✉e-mail: crivelli@phys.ethz.ch

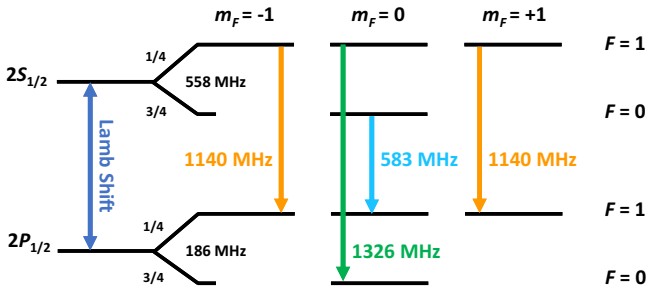

**Fig. 1 | The transitions of the M $2S_{1/2} - 2P_{1/2}$ levels.** The theoretical $2S$ and $2P$ hyperfine splittings are included. $F$ is the quantum number describing the coupling of orbital angular momentum $J$ and nuclear spin $I$, whereas $m_F$ is its magnetic quantum number. The Lamb shift is defined as the difference between the centroid values of the $2S_{1/2}$ and $2P_{1/2}$ levels.

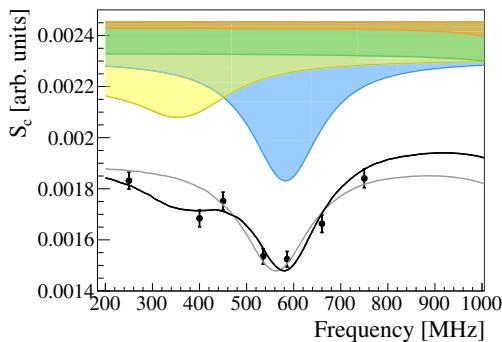

**Fig. 2 | Muonium frequency scan at 22.5 W in the range of 200–800 MHz.** The fitted black line is with, the gray line without the $3S$ contribution. The error bars correspond to the counting statistical error. The colored areas represent the underlying contributions from $2S - 2P_{1/2}$ transitions, namely 583 MHz (blue), 1140 MHz (orange), 1326 MHz (green), and the combined $3S - 3P_{1/2}$ (yellow). The data point with TL OFF is not displayed in the figure, but is included in the fit; it lies at $20.4(4) \times 10^{-4}$.

## Table 1 | Fitting results

| 3S | $S_{BKG}$ ($\times 10^{-4}$) | $B_{2S}$ ($\times 10^{-4}$) | $B_{3S}$ ($\times 10^{-4}$) | $f_{offset}$ (MHz) | $\chi^2_\nu$ |
|---|---|---|---|---|---|
| x | 7.8 (0.8) | 22.0 (1.8) | – | −24.8 (7.4) | 6.7 |
| ✓ | 3.4 (1.1) | 25.0 (1.9) | 5.0 (0.9) | 2.3 (6.8) | 2.0 |

Both fitting results with and without a potential $3S$ contribution are shown. The fitting parameters are explained in "Methods". The $\chi^2_\nu$ corresponds to the reduced chi-squared value with $\nu$ degrees of freedom. $\nu$ is 4 with and 5 without a $3S$ contribution.

## Table 2 | Summary of results

| | Central value | Uncertainty |
|---|---|---|
| Fitting | 580.2 | 6.8 |
| MW-beam alignment | | <0.16 |
| MW field intensity | | <0.07 |
| M velocity distribution | | <0.01 |
| AC Stark $2P_{3/2}$ | +0.39 | <0.02 |
| $2^{nd}$-order Doppler | +0.03 | <0.01 |
| Earth's field | | <0.05 |
| Quantum interference | | <0.04 |
| $2S, F = 0 - 2P_{1/2}, F = 1$ | 580.6 | 6.8 |
| Lamb shift | 1045.5 | 6.8 |
| Theoretical value LS[28] | 1047.498 | 0.001 |
| 2S HFS | 559.6 | 7.2 |
| Theoretical value HFS | 557.9 | <0.1 |

Central values and uncertainty contributions in MHz. The theoretical HFS value was estimated by taking the 1S HFS theoretical value in Muonium[48] and dividing it by a factor of $2^3$. The uncertainty was estimated by calculating the QED $D_{21}$ difference[49].

$F = 1$ is preferred over the other two allowed transitions (see Fig. 1). Due to the reduced influence of other nearby transitions, the systematic uncertainties related to line-pulling would become negligible.

We report here the measurement of the isolated transition between certain hyperfine components of the Muonium the Lamb shift. Combining this result with our previous measurement[22], we determine the $2S$ hyperfine splitting in Muonium. Additionally, we present the observation of Muonium in the $n = 3$ excited state.

## Results

The experimental data and the fits are shown in Fig. 2 and Table 1, respectively. Since for the analysis, the TL OFF data point ($S_c = 20.4(4) \times 10^{-4}$) was set far off-resonance at 2000 MHz, it is not displayed in the figure. When the data are fit without any $3S$ contribution and hence $B_{3S}$ fixed to 0, the reduced $\chi^2$ is 6.7 (5 degrees of freedom) and one obtains a −24.8(7.4) MHz offset compared to the theoretical value. By freeing the $3S$ population parameter, the fit improves to a reduced $\chi^2$ of 2.0 (4 degrees of freedom, $P$ value of 0.09). The frequency offset is found to be 2.3(6.8) MHz. Both fits of the data are shown in Fig. 2, where the gray line corresponds to the fit without and the black line with a $B_{3S}$ contribution. The colored lineshapes represent the underlying transitions with resonances around 583 MHz (blue), 1140 MHz (orange), 1326 MHz (green), and a combined $3S - 3P_{1/2}$ line-shape (yellow).

The main systematic uncertainties are similar to the ones we calculated in ref. 22 and total in 0.19 MHz. The results are summarized in Table 2. The main difference is that the beam contamination in form of $3S$ states is taken into account in the fitting error already. Furthermore, due to their dependence on the resonance frequency, the systematic error stemming from the Doppler shift is approximately halved and the one coming from the uncertainty in the MW field intensity is doubled.

4.2 standard deviations calls for further scrutiny. Muonium (M), the bound state of the positive muon ($\mu^+$) and an electron is an ideal system to study the muon properties and hunt for possible new effects. Due to its lack of sub-structure, it is free from finite nuclear size effects and is therefore an excellent candidate to probe bound-state QED[10], and search for physics beyond the Standard Model[11–13].

Precise measurements of the ground-state hyperfine structure (HFS)[14] and the $1S - 2S$ transition[15] in Muonium were performed, with improvements proposed by MuSEUM (HFS,[16]) and Mu-MASS ($1S - 2S$,[17]) ongoing. All the measurements so far of the M Lamb shift (LS) are limited by statistics[18–20]. Only recently, the formation of an intense metastable M(2S) beam was demonstrated by the Mu-MASS collaboration at the low-energy muon beamline (LEM) at PSI[21], opening up the possibility for precision measurements of the Muonium Lamb shift.

A first measurement at the LEM beamline resulted in a LS value of 1047.2(2.5) MHz[22], representing an order of magnitude improvement upon the previous measurements. With the muCool beamline[23] and (if approved) the high intensity muon beam (HiMB) upgrade at PSI[24], the $\mu^+$ beam quality and flux will further improve, making it feasible to reach statistical uncertainties of the order of a few tens of kHz within a few days of beamtime.

The prospect of not being limited by statistics calls for a systematically more robust method to extract the Lamb shift. Following the example of the most precise measurements of the hydrogen Lamb shift[25–27], extracting the LS from the isolated transition $2S_{1/2}, F = 0 \rightarrow 2P_{1/2},$

**Table 3 | Summary of inputs to the simulation of the lineshapes**

| Initial state | | End state | | Multiplicity | Frequency |
|---|---|---|---|---|---|
| $n^{2S}L_j$ | $(F, m_F)$ | $n^{2S}L_j$ | $(F, m_F)$ | | (MHz) |
| $2^2S_{1/2}$ | (0, 0) | $2^2P_{1/2}$ | (1, 0) | 1 | 582.5 |
| $2^2S_{1/2}$ | (1, 0) | $2^2P_{1/2}$ | (0, 0) | 1 | 1326.4 |
| $2^2S_{1/2}$ | (1, ±1) | $2^2P_{1/2}$ | (1, ±1) | 2 | 1140.5 |
| $3^2S_{1/2}$ | (0, 0) | $3^2P_{1/2}$ | (1, 0) | 1 | 174.1 |
| $3^2S_{1/2}$ | (1, 0) | $3^2P_{1/2}$ | (0, 0) | 1 | 394.5 |
| $3^2S_{1/2}$ | (1, ±1) | $3^2P_{1/2}$ | (1, ±1) | 2 | 339.4 |

A global frequency offset is taken in the fitting. Values for the hyperfine splitting are scaled from the measured HFS of ground state[14] applying corrections given in ref. [50].

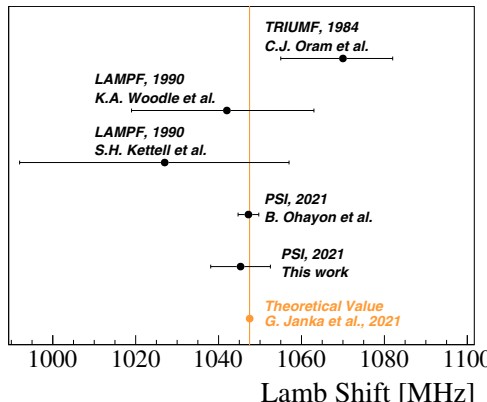

**Fig. 3 | Comparison of theory and experiment.** Summary of all measurements of the $n = 2$ Lamb shift in Muonium (black)[18–20,22] and the latest theoretical value (orange)[28]. The error bars correspond to the uncertainties associated to the experimental or calculated values.

From this measurement, we extract the $2S_{1/2}, F = 0 \rightarrow 2P_{1/2}, F = 1$ resonance frequency to be 580.6(6.8) MHz. Using the calculated values of the $2S$ and $2P$ hyperfine splittings (557.9 MHz and 186.1 MHz, see also Table 3), we determine the M LS to be 1045.5(6.8) MHz. Our result agrees well with the theoretical value of 1047.498 (1) MHz[28] and is limited by statistical uncertainty. A summary of all available measured values of the M LS is shown in Fig. 3. Using our results of the $2S_{1/2}$, $F = 1 \rightarrow 2P_{1/2}, F = 1$ resonance frequency[22], we extract the $2S$ hyperfine splitting in Muonium to be 559.6(7.2) MHz.

The detected $B_{3S}/B_{2S}$ ratio is 0.20(4). In the 60 ns from the foil to the entrance of the detection setup for an average M atom, 32% of the $3S$ states relax back to the ground state. An additional uncertainty in the estimated $3S/2S$ ratio arises from the assumption that the detection efficiency of Ly$\alpha$ and Ly$\beta$ is similar. Those efficiencies depend on a number of factors, which make an accurate determination of its wavelength-dependency challenging. From refs. [29–32], one can evaluate a systematic uncertainty of $0.29 \pm 0.07(\text{stat})^{+0.05}_{-0.09}(\text{syst})$. This agrees with the estimations done by C. Fry[33] of 0.36 and is slightly lower than the estimate of 0.44(4) from a combination of hydrogen population measurements[34,35].

To conclude, we measured the $2S_{1/2}, F = 0 \rightarrow 2P_{1/2}, F = 1$ transition in Muonium. This is the same transition as the one used for the most precise determination of the hydrogen Lamb shift[27]. Since this transition is even more isolated from the others compared to hydrogen, it is the best candidate for future precision LS measurements. By combining the data set with the scan of the 1140 MHz resonance and leaving the $2S$ HFS splitting as a free parameter, we extract the $2S$ hyperfine splitting in Muonium. This measurement, combined with the experimental toolbox developed in this work, is the very first step towards a future high-precision measurement of the Muonium $2S$ HFS, which can be done in a beam as demonstrated in hydrogen[36]. Such a measurement becomes interesting at a challenging precision goal of 63 Hz, corresponding to 1/8th of the uncertainty in the ground state HFS stemming from the muon mass. Furthermore, we detected M(3S). This observation opens up the possibility for microwave spectroscopy experiments with Muonium such as the $n = 3$ Lamb shift or the two-photon transition $3S - 3D_{5/2}$. Both these transitions were measured in hydrogen to a high precision[37,38]. A measurement of the $3S - 3D_{5/2}$ transition would be a unique test of Lorentz- and CPT symmetry in the muonic sector[39].

In our measurement, the comparably large population of $3S$ states distorts the line-shape and introduces line-pulling, which might seem to defeat the purpose of choosing the isolated $n = 2$ transition. This is also supported by the marginal p value obtained in the analysis. However, the $n = 3$ population can be electrically quenched with a weak electrical field, leaving a large fraction of the $n = 2$ population unharmed as shown by measurements from ref. [40]. Extending the beamline to depopulate the $3S$ due to its lifetime is another option, but the strong beam straggling at the foil and resulting diffuse beam would reduce the $2S$ flux significantly. This problem could be overcome by exchanging the carbon foil with only a few layers of graphene (around 1-nm thickness)[41] or moving to a gaseous target. Additionally, with a state-of-the-art apparatus such as muCool to compress and cool the beam, or the HiMB project to increase the $\mu^+$ flux at PSI, the measurements would not be statistically limited anymore. This would allow to probe QED effects such as the Barker–Glover and the nucleus self-energy, which are not accessible in hydrogen yet[28].

## Methods
### Muonium beam formation
The LEM beamline at PSI provides low (1–30 keV) energy muons with a rate of up to 20 kHz using a solid neon moderator[42]. In our experiment, we set the energy of the $\mu^+$ to 7.5 keV in order to maximize the number of M(2S) available for the measurement[21]. The beam transport is optimized with several lenses along the beamline, eventually focusing the muons onto a carbon foil at the entrance of the setup shown in Fig. 4.

In contrast to the LS measurements with protons, the flux of muons is around ten orders of magnitude smaller, which makes it essential to have an efficient background suppression without sacrificing efficiency. Therefore, the incoming muons are tagged on an event-by-event basis to be able to discriminate their times-of-flight (TOFs).

When impinging onto the 10-nm thin carbon foil, a $\mu^+$ releases secondary electrons. These electrons are detected by a microchannel plate (Tag MCP) to give the start signal of the TOF measurement. Upon leaving the foil, the tagged $\mu^+$ has a probability of picking up an electron and form M, primarily in the ground state. From measurements with hydrogen[34,35], 5–10% is expected to be formed in $n = 2$ and 2% in $n = 3$. This expectation was confirmed with a measurement of 11(4)%[21] for M in the metastable state. The M(2S) lifetime is limited by the decay time of the muon itself ($\tau_{M(2S)} = 2.2$ μs). M(3S) is unstable ($\tau_{M(3S)} = 158$ ns) and will relax back to the ground state via the intermediate $2P$ state ($\tau_{M(2P)} = 1.6$ ns).

### Microwave system
The formed beam passes first through a transmission line (TL) which depopulates unwanted hyperfine states (HFS TL). As established by the most precise hydrogen Lamb shift measurements, this reduces the background and at the same time narrows and simplifies the overall line-shape to be fitted. By setting the HFS TL at a frequency of 1140 MHz and with an average power of roughly 29 W, according to our simulation, we depopulate the $F = 1, m_F = ±1$ states by 88% and the $F = 1, m_F = 0$ state by 16%. The transition of interest $2S_{1/2}, F = 0 \rightarrow 2P_{1/2}$, $F = 1$ at 583 MHz remains almost unaffected, where its initial state $F = 0, m_F = 0$ is only depopulated by 4%. With a second TL (Scanner TL) we scan the line-shape of this transition. Seven frequency points are

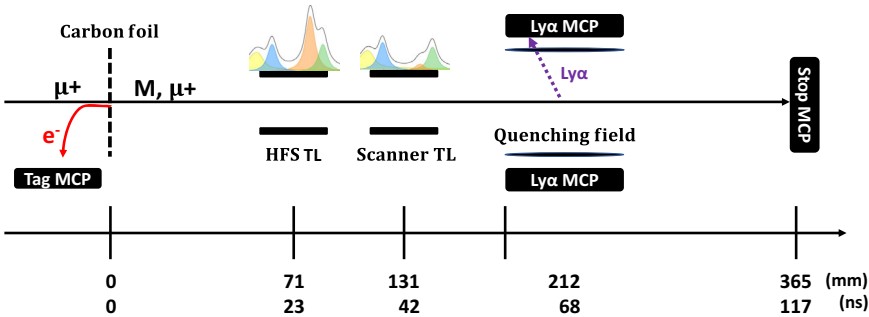

**Fig. 4 | Schematic view of the experimental setup.** The signal signature consists of a coincidence signal between Tag, Lyα and Stop-MCP. The colored lineshapes above the transmission lines (TLs) correspond to the observable transitions (1326 MHz (green), 1140 MHz (orange), 583 MHz (blue), and the combined $3S_{1/2} - 3P_{1/2}$ (yellow) contribution). The time scale is given for the most-probable Muonium atom with an energy of 5.7 keV.

measured in the range of 200–800 MHz with an average power of 22.3 W. An additional data point is taken with the Scanner TL turned off. The output powers of both TLs are continuously monitored with 50-Ohm terminated power meters to ensure the stability of the measurement. We then apply a small correction originating from the measured frequency-dependent power loss from the center of the TL to the power meter. The specific frequency points, as well as the corrected average power ⌀P for each frequency, are summarized in Table 4.

## Detection system

Due to the short lifetime of the 2P states, the atoms excited by the microwave relax to the ground state before reaching the detection setup. The remaining excited states are quenched by applying an electrical field of about 250 V cm⁻¹ between the two grids mounted in front of the coated Lyα-MCPs (see Fig. 4) and relax back to the ground state with the emission of UV photons (2P Lyα: 121.5 nm, 3P Lyβ: 102.5 nm). These photons interact with the coating and release single electrons, which are in turn detected by the Lyα-MCPs. The beam then continues onto the Stop-MCP, giving the stop time for the TOF measurement. An event is recorded only if a coincidence signal between the Tag- and Stop-MCP is seen above threshold (double coincidence). The time window for a valid event is set to 10 μs and multiple hits in the detectors are recorded.

## Analysis

The signature for the detection of M(2S) atoms is defined as the double coincidence and additionally the Lyα-MCP signal in the time region of interest (triple coincidence, see also Fig. 4).

### Table 4 | Summary of data collection

| Frequency (MHz) | $S_{Lyα}$ | $S_{Norm}$ | Time (h) | ⌀P (W) |
|---|---|---|---|---|
| 250 | 3090 | 1,688,525 | 5.3 | 22.04 (4) |
| 400 | 2750 | 1,636,194 | 5.2 | 22.25 (6) |
| 450 | 2843 | 1,627,631 | 5.0 | 22.17 (11) |
| 535 | 2775 | 1,807,147 | 5.7 | 22.45 (17) |
| 586 | 2733 | 1,791,790 | 5.5 | 22.50 (17) |
| 660 | 2844 | 1,712,871 | 5.3 | 22.35 (19) |
| 750 | 2963 | 1,609,137 | 5.3 | 22.09 (20) |
| OFF | 3344 | 1,636,905 | 5.2 | – |
| Total | 23,342 | 13,510,200 | 42.5 | |
| Rate | 0.33 Hz | 88 Hz | | |

The statistics gathered during the measurement as well as rates and average monitored power ⌀P are shown. The rate is the maximal rate, which is calculated by taking only the data with TL OFF and additionally corrected for the active HFS TL.

In Fig. 5, the measured TOFs between Tag- and Stop-MCP (*x* axis) are correlated with the TOFs between Tag- and Lyα-MCP (*y* axis). The energy distribution of the M atoms after the carbon foil is well-known from previous measurements at the LEM beamline and reproduced by simulation[43]. The most-probable energy of the $μ^+$ after the foil is 5.7(2) keV. The expected TOF of an M atom from the Tag- to the Stop-MCP ranges from 90 ns to 135 ns, with a most-probable TOF of 117 ns. Applying this time cut, we extract the detected amount of M and $μ^+$, which serves as normalization factor $S_{Norm} = N_M + N_{μ^+}$, accounting for variations in the beam intensity. The TOF between the tagging and the emission of a Lyα photon is calculated to be in the range of 30 ns to 75 ns. Applying both of these cuts, we identify our signal region, portrayed in Fig. 5. The amount of signal events are extracted for each frequency point $(S_{Lyα}(f))$. The normalized signals per frequency point are then calculated as $S(f) = S_{Lyα}(f)/S_{Norm}$. The summary of the statistics gathered and the rates achieved is shown in Table 4.

In region A of Fig. 5, events with longer TOFs than expected are summarized. Those events might be caused by either convoy electrons created at the foil[44] or by ionization products of a $μ^+$ interacting with the residual gas. The region A is leaking into the signal region, contributing to the measured background. The region B contains the events where the TOFs between the Lyα- and Stop-MCP are almost zero. These MCPs are close to each other, which leads to the risk of one detector picking up a noise signal when the other one is firing or vice versa. This noise signal can have a large amplitude with an intense after-ringing, which is mistaken as an additional signal that contributes to the events in region C.

The average power inside the TLs is frequency-dependent as seen in Table 4. To be able to fit a line-shape, all frequency points are corrected to the same global power of 22.5 W, obtaining the corrected signal $S_c$. The applied correction is given by:

$$S_c(f) = (S(f) - S_{BKG}) \cdot C(f) + S_{BKG},    (1)$$

where $S_{BKG}$ is the background level. The correction factors $C(f)$ are extracted by simulating the scaling ratio between the excitation probabilities at the effectively measured power and the global power for each frequency.

## Fitting procedure

Because of the linear polarization of the field in the TLs, we can approximate our system to two levels and use the relevant optical Bloch equations[45] in the simulation. Single-atom trajectories through both TLs reaching the Stop-MCP are simulated. While being in one of the TLs, the optical Bloch equations are numerically solved by an adaptive stepsize Runge–Kutta algorithm[46]. To reproduce the non-uniform field inside the TLs, for each TL a field map is generated in

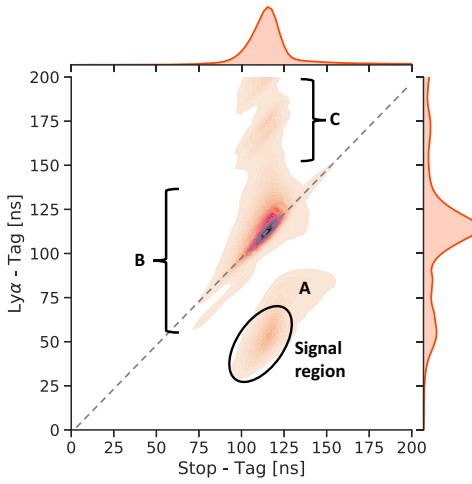

**Fig. 5 | Time-of-flight distributions.** Correlation plot of the TOF between the Tag- and Stop-MCP and the TOF between the Tag- and Lyα-MCP. The signal region of Lyα events is circled. The regions labeled A, B, C are explained in the main text.

SIMION, following the procedure of Lundeen and Pipkin[26]. The numbers of excited and ground-state atoms detected in the Stop-MCP are counted and from that the average excitation probability calculated. As input parameter to the simulation we assign to each atom an initial state with its specific resonance frequency from Table 3, and the power for both TLs operating at random phases of the fields. The momentum and initial position of the specific particle from the LEM is simulated with the musrSIM package[47] beforehand, from which the simulation randomly draws an atom.

The individual line-shape $P^{(i)}$, where $i$ stands for the assigned resonance frequency in MHz, is constructed by simulating the excitation probability for each initial state with large statistics over a frequency range between 200 MHz and 2000 MHz in steps of 1 MHz. Combining all relevant transitions, a global line-shape $P_n$ for a $n$ state is obtained:

$$P_{n=2} = 0.5 \cdot P^{(1140)} + 0.25 \cdot P^{(583)} + 0.25 \cdot P^{(1326)} \tag{2}$$

$$P_{n=3} = 0.5 \cdot P^{(339)} + 0.25 \cdot P^{(174)} + 0.25 \cdot P^{(394)}, \tag{3}$$

where the constant factors are coming from spin statistics.

The fitting function is constructed with the simulated lineshapes:

$$S_c(f) = S_{BKG} + \sum_n \left[ B_n \cdot P_n(f - f_{offset}) \right], \tag{4}$$

where $B_n$ is scaling the line-shape $P_n$ for a specific $n$ state and $f_{offset}$ is introduced to allow for a global frequency offset compared to the simulated resonance frequencies.

## Data availability
The data that support the plots within this paper and other findings of this study are available from the corresponding author upon request.

## Code availability
The simulation and the data analysis are developed using custom codes based on publicly available frameworks: geant4 (https://geant4.web.cern.ch/) and ROOT. The codes are available from the corresponding author upon request.

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

## Acknowledgements
This work is based on experiments performed at the Swiss Muon Source SµS, Paul Scherrer Institute, Villigen, Switzerland. This work is supported by the ERC consolidator grant 818053-Mu-MASS (P.C.) and the Swiss National Science Foundation under grant 197346 (PC). B.O. acknowledges support from the European Union's Horizon 2020 research and innovation program under the Marie Skłodowska-Curie grant agreement No. 101019414, as well as ETH Zurich through a Career Seed Grant SEED-09 20-1.

## Author contributions
G.J., B.O., I.C., Z.B., L.S.B., E.D., A.G., X.N., Z.S. A.S., T.P., and P.C. have contributed to this publication, being variously involved in the design and construction of the detectors, writing software, calibrating subsystems, operating the detectors, acquiring data, and analyzing the processed data.

## Competing interests
The authors declare no competing interests.
