## [Peer Review File · Nature Communications]

Measurement of the transition frequency from $2S_{1/2}, F=0$ to $2P_{1/2}, F=1$ states in MuoniumREVIEWER COMMENTS

Reviewer #1 (Remarks to the Author):

Referee report on „Measurement of the $2S_{1/2}, F=0 \rightarrow 2P_{1/2}, F=1$ transition in Muonium”

by G. Janka, et al.

This is a very well written paper, in an interesting style. My doubts are regarding the scientific contents and I will omit the „Experimental method” section. Authors have previously [20] measured $2S_{1/2}, F=1 \rightarrow 2P_{1/2}, F=1$ transition frequency and determined Lamb shift. In this work, they measure $2S_{1/2}, F=0 \rightarrow 2P_{1/2}, F=1$ and determine Lamb Shift with 2.5 times worse accuracy. So, from the theoretical point of view only, it is not a significant achievement. So, I would expect some explanations, what is really new here. Apart of this main point I have several minor comments listed below.

1: What is Authors’ definition of the Lamb shift ?, this should be explicitly written somewhere, Namely it is the difference in the centroid values of $2S_{1/2}$ and $2P_{1/2}$ levels. This is important in a view of significant mixing between $P_{1/2}$ and $P_{3/2}$ states.

2: To obtain Lamb shift, Authors most probably, subtract theoretical values for the hyperfine of $2S_{1/2}$ and $2P_{1/2}$. This should be explicitly written, and hyperfine values given. Later Authors determine hfs of $2S_{1/2}$, which they do not use to get the Lamb shift. They prefer theoretical predictions much more, what is not surprising.

3: The value of their measured frequency 580.6(68) MHz does not somehow coincide with the value given on Fig. (1) 583 MHz, why ?

4: Authors write in Abstract: „ ...which might provide an explanation of the g-2 muon anomaly or Lorentz and CPT violation. „. While muon g-2 is a well established anomaly, it is not the case of Lorentz and CPT violation. I would not put them on the same footing.

Regarding final recommendations, I am not able to give at this point, will have to see Authors’ respond.

With best regards

Krzysztof Pachucki

Reviewer #2 (Remarks to the Author):

[General comments]

In this paper, the frequency of a previously unmeasured transition has been determined by in-flight microwave spectroscopy of muonium atoms. Spectroscopy of muonium helps to validate the Standard Model, including bound-state QED, with high precision. At the same time, it enables searches for physics beyond the Standard Model of particle physics.

The key outcomes in this paper are the first determination of the 2S-HFS, the independent determination of the $n=2$ Lamb shift, and the observation of the $n=3$ muonium. All of them are important results that will impact the future of the research field. On the other hand, it is somewhat controversial whether the paper sufficiently conveys the novelty and significance to the broad readership of Nature Communications (Even the journal's goal is to report on the critical advances of significance to specialists within each field). Since the text is only about 3300 words compared to the journal's regulation of 5000 words, the reviewer believes that a more specific and detailed description of the experimental method and data analysis should be added to the text.

This paper describes results following the $n=2$ Lamb shift spectroscopy of muonium reported by the group at Physical Review Letters this year (B. Ohayon et al., "Precision Measurement of the Lamb Shift in Muonium," Phys. Rev. Lett., 128, 011802 (2022), hereinafter referred to as Oyahon2022.). It is possible that this paper could be interpreted as an independent confirmation of the results reported in Oyahon2022, in which case it is difficult to determine from this manuscript whether the level of novelty and significance required by the journal has been satisfied.

Another minor concern is that the uncertainty notation of the measurement result is inconsistent with that of Oyahon2022. For example, the Lamb shift is shown as $1047.2(2.3)_{\text{stat}}(1.1)_{\text{syst}}$ MHz in Oyahon2022, which is correct. However, this paper notes $1047.2(25)$ MHz, with the decimal point in the incorrect position. This error is also found in other quantities such as 2S-HFS, so the reviewer recommends that the authors check the manuscript again.

From the reviewer's point of view, the scientific significance of the motivation and the importance of the results stated in this paper are sufficient. So the reviewer would like to see a more detailed and substantive description enhanced.

[Comments on each section]

0. Abstract

0-1: The mention of the muon discovery and g-2 deserves an introduction rather than an abstract.

1. Introduction

1-1: The reviewer finds the first three paragraphs somewhat essayistic, not to say unnecessary, but not deeply relevant enough to be the basis for later discussion. The reviewer suggests compressing the descriptions.

1-2: Does the precision measurement of 2S-HFS have unique significance compared to 1S-HFS? If so, it is an important feature and should be explained.

1-3 The reviewer understands that the Barker-Glover effect is part of the recoil correction, but what is the significance of being able to verify this as a test for bound-state QED?

2. Experiment

2-1: It may be semi-obvious, but F and mF should be defined in the text.

2-2: On what basis was the microwave power in the transmission line set? Is it possible to make quantitative mentions of the uniformity of the power in the area through which the muonium passes and the stability of the power during the measurement?

2-3: The phrase "an electrical field of the order of 250 Vcm⁻¹" in the fourth paragraph seems a bit odd (too specific to be "an order"). Why not simply use about 250 Vcm⁻¹ here?

2-4: The reviewer recommends that $Ly\alpha$ and $Ly\alpha$ be unified since they are mixed in the figures and the text.

2-5: If the events in region A are derived from convoy electrons or ionization products, can we distinguish them from the signal by using the difference in waveform or pulse height of the MCP signals?

2-6: As shown in region C, does only $Ly\alpha$ -MCP has ringing in the analog pulse and Stop-MCP has no ringing?

2-7: How was the microwave power measured? If the power was picked up using an antenna, then the antenna's frequency response must be considered. Or was the power measured with a transmitting or terminated power sensor?

2-8: Why is the correction factor $C(f)$ determined by the effectively measured power even though microwave power is independent of frequency?

2-9: What is the origin of the frequency offset (f_{offset})? Is this a source of systematic error?

3. Results

3-1: Reduced chi-square = 2.0 has a different meaning depending on the degree of freedom. For example, $p=0.08$ is obtained for chi-square 10 with 5 degrees of freedom, $p=0.03$ for chi-square 20 with 10 degrees of freedom. To eliminate ambiguity, the reviewer recommends describing both chi-square and degrees of freedom instead of reduced chi-square.

3-2: Reduced chi-square=2 seems to imply incompleteness of the fitting function. Are there any candidates for further corrections that should be considered?

3-3: How was the TL-off data point in the line-shape included in the fitting? What does this mean in line-shape analysis (estimation of S_{BKG})?

3-4: How accurate is the assumption that Ly α and Ly β have similar detection efficiency? Is it possible to discuss the degree of uncertainty, even if it is difficult to evaluate precisely?

4. Conclusions

4-1: Does this comparison with the hydrogen spectroscopy results provide any new insights? If so, it would be good to be a little more specific.

4-2: Is there a possibility that n=3 spectroscopy, such as 3S-3D $5/2$, can be performed with muonium to gain new insights that have not been obtained before? If so, it would be good to be a little more specific.

Reviewer #3 (Remarks to the Author):

The paper present an important result in the field that may be of interest of a relatively broad audience.

1). In the introduction g-2 of muon is mentioned. It is said about a 4.2 sigma deviation from a prediction. A theoretical reference is required. It is said that the deviation is 'confirmed'. It would be good to mention what result was confirmed with a reference.

2). I do not completely understand the wording about 'finite size'. In a sense a muon or an electron have finite-size effects, but they are calculable. The Lamb shift is a result of the size of an electron. The problem is the nuclear structure (of nuclei heavier than a proton) or the proton structure. They are not calculable and they include the nuclear finite size and nuclear polarizability. One should rather speak about hadronic structure, hadronic effects, or effects of strong interactions etc.

3). To my taste, there are too many abbreviations in general and per a sentence.

4). Term 'hyperfine transition(s)' is confusing. A transition between certain [hyperfine] components of the Lamb shift is not a hyperfine transition.

5). 'Results' section: It is said that reduced $\chi^2=6.7$ and 2 (for different fits). 6.7 is definitely bad and that is why the other fir was performed. But I do not understand whether 2 is OK. It would be good to

say what is the number of degrees of freedom and what is the probability that reduced χ^2 for those dof is >2.0 .

6). Paragraph below Fig. 4: A theoretical value is mentioned. Then an appropriate reference should be given there.

Most of my comments are minor. The exception is the \#5 one. I cannot understand how good or bad is the fit. The paper probably deserves to be published. However, the issue on the quality of the fit should be clarified first.

Dear Editor,

We would like to thank you for your kind cooperation in the review process of our manuscript. We are also thankful to the referees for spending their valuable time to review our manuscript. Our response (in blue) to the Referees' comments and questions are listed below, accompanied with a full list of modifications made to the manuscript at the end of this letter.

Referee A:

1. This is a very well written paper, in an interesting style. My doubts are regarding the scientific contents and I will omit the "Experimental method" section. The authors have previously measured $2S_{1/2}, F = 1 - 2P_{1/2}, F = 1$ transition frequency and determined Lamb shift. In this work, they measure $2S_{1/2}, F = 0 - 2P_{1/2}, F = 1$ and determine Lamb Shift with 2.5 times worse accuracy. So, from the theoretical point of view only, it is not a significant achievement. So, I would expect some explanations, what is really new here.

Dear Krzysztof,

we thank you for your review and understand your concerns that from the theoretical point of view a confirmation of our previous results is not a major achievement. The novelty of this work is experimental. We measure for the first time the $2S_{1/2}, F = 0 - 2P_{1/2}, F = 1$ transition in muonium. This is not just another line, which measurement for exotic atoms such as muonium is anyway much more challenging than in ordinary atoms, but is the one isolated from the others. Learning from what was achieved in hydrogen, this opens up the possibility for greatly improved measurements of the muonium Lamb shift. Moreover, we determine for the first time the HFS $n = 2$. Even though it is far from being competitive with the ground state HFS measurement, it is the first step towards a dedicated high-precision measurement. This point is now emphasized in our conclusions (See correction 23 in the list below).

By looking at publications from our colleagues in the exotic atom community we think that Nature Communications is an appropriate journal to report important advancements in the field which, even though not immediately applicable to improve our knowledge from the theoretical side, open new exciting prospects for the near future.

Below are some examples:

"Sympathetic cooling of positrons to cryogenic temperatures for antihydrogen production", Nature Communications 12, (2021) 6139.

"Pulsed production of antihydrogen", Nature Communications 4, (2021) 19.

"Antihydrogen accumulation for fundamental symmetry tests", Nature Communications 8, (2017) 681.

"A moiré deflectometer for antimatter", Nature Communications 5, (2014) 4538.

"A Source of antihydrogen for in-flight Hyperfine spectroscopy", Nature Communications 5, (2014) 3089.

"Description and first application of a new technique to measure the gravitational mass of antihydrogen", Nature Communications 4, (2013) 1785.

Apart of this main point I have several minor comments listed below:

2. What is Authors' definition of the Lamb shift? This should be explicitly written somewhere, Namely it is the difference in the centroid values of $2S_{1/2}$ and $2P_{1/2}$ levels. This is important in a view of significant mixing between $P_{1/2}$ and $P_{3/2}$ states.

We now specify explicitly. See Corr. 9.

3. To obtain the Lamb shift, the Authors most probably subtract theoretical values for the hyperfine of $2S_{1/2}$ and $2P_{1/2}$. This should be explicitly written, and hyperfine values given. Later Authors determine HFS of $2S_{1/2}$, which they do not use to get the Lamb shift. They prefer theoretical predictions much more, what is not surprising.

Thanks for pointing this out. We added this detail to both the main text and table 2. We also added detail and reference for our estimation of the 2S and 2P HFS values. See Corr. 14, 20.

4. The value of their measured frequency 580.6(6.8) MHz does not somehow coincide with the value given on Fig. (1) 583 MHz, why?

In Fig. 1 are the theoretical predictions. Our measured frequency coincides with the theoretical prediction within 1 sigma. We made several changes to emphasize this. See Corr. 9, 14, 20.

5. Authors write in Abstract: "... which might provide an explanation of the g-2 muon anomaly or Lorentz and CPT violation." While muon g-2 is a well established anomaly, it is not the case of Lorentz and CPT violation. I would not put them on the same footing.

Thank you for pointing out this mistake. We rewrote this sentence. See Corr. 5.

Regarding final recommendations, I am not able to give at this point, will have to see Authors' response.

Referee B:

1. In this paper, the frequency of a previously unmeasured transition has been determined by in-flight microwave spectroscopy of muonium atoms. Spectroscopy of muonium helps to validate the Standard Model, including bound-state QED, with high precision. At the same time, it enables searches for physics beyond the Standard Model of particle physics.

The key outcomes in this paper are the first determination of the 2S-HFS, the independent determination of the $n = 2$ Lamb shift, and the observation of the $n = 3$ muonium. All of them are important results that will impact the future of the research field. On the other hand, it is somewhat controversial whether the paper sufficiently conveys the novelty and significance to the broad readership of Nature Communications (Even the journal's goal is to report on the critical advances of significance to specialists within each field). Since the text is only about 3300 words compared to the journal's regulation of 5000 words, the reviewer believes that a more specific and detailed description of the experimental method and data analysis should be added to the text.

This paper describes results following the $n = 2$ Lamb shift spectroscopy of muonium reported by the group at Physical Review Letters this year [...]. It is possible that this paper could be interpreted as an independent confirmation of the results reported in Ohayon2022, in which case it is difficult to determine from this manuscript whether the level of novelty and significance required by the journal has been satisfied. [...] From the reviewer's point of view, the scientific significance of the motivation and the importance of the results stated in this paper are sufficient. So the reviewer would like to see a more detailed and substantive description enhanced.

We thank the reviewer for the positive assessment and the valuable suggestions to enhance and substantiate the description. As suggested, we added more details to the manuscript. Namely: we made the definitions of the LS and the interplay with HFS theory more explicit (Corr. 9, 14, 20), added more experimental details (Corr. 10, 11, 13), added more details on the fitting and analysis (Corr. 14, 16, 15, 17, 18, 19), and added more details to the conclusions (Corr. 22, 23, 24)

2. Another minor concern is that the uncertainty notation of the measurement result is inconsistent with that of Ohayon2022. For example, the Lamb shift is shown as 1047.2(2.3)stat(1.1)syst MHz in Ohayon2022, which is correct. However, this paper notes 1047.2(25) MHz, with the decimal point in the incorrect position. This error is also found in other quantities such as 2S-HFS, so the reviewer recommends that the authors check the manuscript again.

The notation was changed. See Corr. 1.

3. The mention of the muon discovery and g-2 deserves an introduction rather than an abstract.

Comparing with e.g. ref (1) in the list above, we thought that this is the style supported by the journal. If this is also the editor recommendation, we would remove this part from the abstract and only keep it in the introduction.

4. The reviewer finds the first three paragraphs somewhat essayistic, not to say unnecessary, but not deeply relevant enough to be the basis for later discussion. The reviewer suggests compressing the descriptions.

We were under the impression that such an introduction fits with the style of this journal. Nevertheless, we compressed the introduction slightly. See Corr. 6

5. Does the precision measurement of 2S-HFS have unique significance compared to 1S-HFS? If so, it is an important feature and should be explained.

Currently, the theory of the 1S HFS in muonium is dominated by our knowledge of the muon mass so that the much better experimental accuracy is not exploited for testing QED or new physics searches. We added in the conclusion the challenging precision goal for which a 2S HFS measurement would be useful. See Corr. 23.

6. The reviewer understands that the Barker-Glover effect is part of the recoil correction, but what is the significance of being able to verify this as a test for bound-state QED?

In light atoms where one constituent is much heavier than the other, bs-QED is developed as a series in m/M , α , αZ , and nuclear quantities. Different atoms and transitions probe different parts of this parameter space. The Barker-Glover term is an example of a correction which is second order in recoil, so it is two orders of magnitude larger in muonium than in hydrogen, where it is negligible.

7. It may be semi-obvious, but F and mF should be defined in the text.

They are now defined in the text as well. See Corr. 9.

8. On what basis was the microwave power in the transmission line set? Is it possible to make quantitative mentions of the uniformity of the power in the area through which the muonium passes and the stability of the power during the measurement?

The input power in the transmission line was set so that we measure the same output power for each frequency. We then apply a small correction originating from the measured frequency-dependent power loss from the center of the TL to the power meter. This corrected power is summarized in Tab. 1 together with the standard deviation indicated (Corr. 13). We also added this info to the text (Corr. 11).

We don't assume that the microwave field is uniform, but simulate a field map with SIMION and include it in the lineshape simulation. This was added in the text. See Corr. 15

9. The phrase "an electrical field of the order of 250 V cm^{-1} " in the fourth paragraph seems a bit odd (too specific to be "an order"). Why not simply use about 250 V cm^{-1} here?

Changed as suggested. See Corr. 12

10. The reviewer recommends that $Ly\alpha$ and $Ly\alpha$ be unified since they are mixed in the figures and the text.

Unified to $Ly\alpha$. See Corr. 2

11. If the events in region A are derived from convoy electrons or ionization products, can we distinguish them from the signal by using the difference in waveform or pulse height of the MCP signals?

Thank you for the very useful suggestion. We did not record wave-forms in this work. We are now considering the use of digitizers so we will be able to test this in future work.

12. As shown in region C, does only Lya-MCP has ringing in the analog pulse and Stop-MCP has no ringing?

The way we read out the stop-MCP precluded us from seeing multi-hits on it, so that only the leading edge was recorded without the ringing.

13. How was the microwave power measured? If the power was picked up using an antenna, then the antenna's frequency response must be considered. Or was the power measured with a transmitting or terminated power sensor?

The power was measured with a calibrated, terminated power sensor and an attenuator. We now added this to the text. See Corr. 10.

14. Why is the correction factor $C(f)$ determined by the effectively measured power even though microwave power is independent of frequency?

The average power inside the TLs is frequency-dependent. The correction factor $C(f)$ is determined to allow us to correct the individual frequency points to a common power. We added a new table (Tab. I) with the average power and its standard deviation for each frequency. See Corr. 13.

15. What is the origin of the frequency offset (f_{offset})? Is this a source of systematic error?

f_{offset} is a parameter that accounts for a global frequency deviation of the fit from the simulated resonance positions, for which theoretical values are used. f_{offset} plus the corrections in table III are equal to the deviation between experiment and theory. We added more info on f_{offset} to the text. See Corr. 14, 17

16. Reduced chi-square = 2.0 has a different meaning depending on the degree of freedom. For example, $p=0.08$ is obtained for chi-square 10 with 5 degrees of freedom, $p=0.03$ for chi-square 20 with 10 degrees of freedom. To eliminate ambiguity, the reviewer recommends describing both chi-square and degrees of freedom instead of reduced chi-square.

The number of degrees of freedom as well as the p-value was added. See Corr. 18.

17. Reduced chi-square=2 seems to imply incompleteness of the fitting function. Are there any candidates for further corrections that should be considered?

In our fitting function we only included $2S - 2P_{1/2}$ and $3S - 3P_{1/2}$ transitions. They are now added table II 14 to make it clearer how the fitting function was formed. Including also $J = 3/2, 3D$ states and higher excited transitions might improve the fit. However, in future $n = 2$ Lamb shift measurements, the aim would be to depopulate $n \leq 3$ states with an electrical field to avoid such complications. We added a reference to the marginal chi2 in the conclusion, see Corr. 22.

18. How was the TL-off data point in the line-shape included in the fitting? What does this mean in line-shape analysis (estimation of S_{BKG})?

The TL-off data point was included in the line-shape as a point far off-resonance at 2000 MHz. We now added it to the caption of Figure 4, and the first paragraph of the results. See Corr. 19.

19. How accurate is the assumption that Lya and Lyb have similar detection efficiency? Is it possible to discuss the degree of uncertainty, even if it is difficult to evaluate precisely?

We added an estimation of the uncertainty and added this as an additional systematic uncertainty. We also added two more references [42,43]. See Corr. 21.

20. Does this comparison with the hydrogen spectroscopy results provide any new insights? If so, it would be good to be a little more specific.

Since muonium is very similar to hydrogen, a lot can be learned and adapted from the hydrogen measurements, e.g. ways to cross-check systematics, line-shape modelling etc. This becomes important as muonium measurements advance to the point where they are not limited by statistics.

21. Is there a possibility that $n = 3$ spectroscopy, such as $3S - 3D_{5/2}$, can be performed with muonium to gain new insights that have not been obtained before? If so, it would be good to be a little more specific.

Transitions involving j different angular momenta probe different parts of the the Lorenz and CPT violation parameter space. We added a sentence about the impact of a measurement of the $3S - 3D_{5/2}$ transition in Muonium and an additional reference. See Corr. 24

Referee C:

1. The paper present an important result in the field that may be of interest of a relatively broad audience.

We thank you for reviewing our manuscript and your positive assessment.

2. In the introduction g-2 of muon is mentioned. It is said about a 4.2 sigma deviation from a prediction. A theoretical reference is required. It is said that the deviation is ‘confirmed’. It would be good to mention what result was confirmed with a reference.

The proper citations were added. See Corr. 8.

3. I do not completely understand the wording about ‘finite size’. In a sense a muon or an electron have finite-size effects, but they are calculable. The Lamb shift is a result of the size of an electron. The problem is the nuclear structure (of nuclei heavier than a proton) or the proton structure. They are not calculable and they include the nuclear finite size and nuclear polarizability. One should rather speak about hadronic structure, hadronic effects, or effects of strong interactions etc.

We revised to “finite nuclear size”. See Corr. 7.

4. To my taste, there are too many abbreviations in general and per a sentence.

The amount of abbreviations was reduced. See Corr. 3.

5. Term ‘hyperfine transition(s)’ is confusing. A transition between certain [hyperfine] components of the Lamb shift is not a hyperfine transition.

This was indeed confusing and we corrected as suggested. See Corr. 4.

6. ‘Results’ section: It is said that reduced $\chi^2 = 6.7$ and 2 (for different fits). 6.7 is definitely bad and that is why the other fir was performed. But I do not understand whether 2 is OK. It would be good to say what is the number of degrees of freedom and what is the probability that reduced χ^2 for those dof is > 2.0 .

The number of degrees of freedom and the probability which is $p=0.09$ were added to the text. Please refer to our reply to reviewer 2 for more details on why the χ^2 is marginal. See Corr. 18 22.

7. Paragraph below Fig. 4: A theoretical value is mentioned. Then an appropriate reference should be given there.

Thank you this was added. See Corr. 20.

Most of my comments are minor. The exception is the 5th one. I cannot understand how good or bad is the fit. The paper probably deserves to be published. However, the issue on the quality of the fit should be clarified first.

List of modifications made to the manuscript

1. General: Notation of uncertainties corrected and made consistent with Ohayon2022.
2. General: Unified $Ly\alpha$ and $Ly\alpha$ in all cases to $Ly\alpha$ (text, Fig.2, Fig.3).
3. General: Amount of abbreviations (especially M LS or H LS) was reduced.
4. General: We revised the hyperfine transition according to context
5. Abstract: We corrected the sentence about g-2 and Lorentz and CPT.
6. Intro, 1st and 2nd paragraphs: Compressed the text.
7. Intro, 2nd paragraph: changed from “finite size” to “finite nuclear size”.
8. Intro, 3rd paragraph: Added citations for g-2 theory [9] as well as BNL [8] measurement.
9. Experiment, Fig 1: Added definition of Lamb shift. Noted that the hyperfine splittings are theoretical. Added definition of quantum numbers F and mF. Removed the word ”hyperfine” from ”hyperfine transitions”.
10. Experiment, 4th paragraph: added the word “50-Ohm terminated” for the power meter description.
11. Experiment, 4th paragraph: added the sentence ”We then apply a small correction originating from the measured frequency-dependent power loss from the center of the TL to the power meter. The specific frequency points as well as the average power \bar{P} for each frequency are summarized in Tab. 1.”
12. Experiment, 5th paragraph: revised the sentence about the electrical field of 250V/cm.
13. Experiment, Added table I, with measured frequencies, measurement times, achieved rates and average monitored power. Added references to the table in the main text.
14. Experiment, added Table II, with the simulated data used for the analysis. In the caption we explain about the global frequency offset, and note that how the hyperfine splittings are calculated.
15. Experiment, 8th paragraph: Extended sentence to ”To reproduce the non-uniform field inside the TLs, for each TL a field map is generated in SIMION, following the procedure of Lundeen and Pipkin”
16. Experiment, revised equation 4.
17. Experiment, last sentence. Added explanation on f_{offset} .
18. Results, 1st paragraph: written explicitly the number of degrees of freedom and p-value for the fits.
19. Results, Fig 4, and 1st paragraph: added sentence explaining how TL OFF point was incorporated into the analysis.
20. Results, 3rd paragraph: added information on the theoretical values of the HFS and LS.
21. Results, 4th paragraph: added uncertainty estimation from $Ly\alpha$ and $Ly\beta$ efficiencies and added two more refs [42,43] and revised sentence with result.
22. Conclusions: added ”This is also supported by the marginal p-value obtained in the analysis”
23. Conclusions: added a paragraph on the prospects of a high-precision measurement of the 2S HFS.
24. Conclusions: added sentence about $3S - 3D_{5/2}$ transition and interest for Lorentz- and CPT tests with additional citation [47].

REVIEWERS' COMMENTS

Reviewer #1 (Remarks to the Author):

Authors have responded to all my queries and presented convincing arguments, so I can now recommend this paper for the publication, with two minor comments:

1: I would not use the word „fascinating” in the Abstract, and maybe also in the main paper.

2: Authors write the same sentence twice: „This would allow to probe QED effects such as the Barker-Glover and the nucleus self-energy, which are not accessible in hydrogen yet [25]”, I would suggest to avoid this and change the second one.

With best regards

Krzysztof Pachucki

Reviewer #2 (Remarks to the Author):

The manuscript was revised in response to the reviewers' comments. The reviewer highly appreciates that the revisions have improved the clarity and accuracy of the descriptions. In particular, additional details on experimental methods, analysis, and evaluation of systematic uncertainties are significant. The authors addressed each of the points noted and answered the concerns and questions stated in the first review.

Muonium spectroscopy plays a unique role in testing the Standard Model and searching for new physics. As noted in the manuscript, it can also provide insight not available from studies using ordinary atoms. The experimental method is new, though still being developed, and may lead to future breakthroughs in the research area. The paper's contents will not only appeal to a limited field but will also interest a wide range of readers.

In conclusion, the reviewer recommends that this paper be published.

Reviewer #3 (Remarks to the Author):

I am satisfied with the changes and recommend to publish the paper as it is.

Dear Editor,

We again would like to thank you and the referees for spending their valuable time to review our manuscript and are delighted that you found our manuscript to be suitable for publishing in Nature Communications. Our response (in blue) to the Referees' comments and questions are listed below.

Referee A:

Authors have responded to all my queries and presented convincing arguments, so I can now recommend this paper for the publication, with two minor comments:

I would not use the word "fascinating" in the Abstract, and maybe also in the main paper.

We removed the word "fascinating" in the abstract.

Authors write the same sentence twice: "This would allow to probe QED effects such as the Barker-Glover and the nucleus self-energy, which are not accessible in hydrogen yet [25]", I would suggest to avoid this and change the second one.

We followed your suggestion and removed the second appearance of the sentence.

With best regards Krzysztof Pachucki

Dear Krzysztof,

Thanks a lot for taking time to your review and very useful comments which allow us to improve our manuscript.

Referee B:

The manuscript was revised in response to the reviewers' comments. The reviewer highly appreciates that the revisions have improved the clarity and accuracy of the descriptions. In particular, additional details on experimental methods, analysis, and evaluation of systematic uncertainties are significant. The authors addressed each of the points noted and answered the concerns and questions stated in the first review.

Muonium spectroscopy plays a unique role in testing the Standard Model and searching for new physics. As noted in the manuscript, it can also provide insight not available from studies using ordinary atoms. The experimental method is new, though still being developed, and may lead to future breakthroughs in the research area. The paper's contents will not only appeal to a limited field but will also interest a wide range of readers.

In conclusion, the reviewer recommends that this paper be published.

We thank the reviewer for the very helpful comments which allowed us to improve our manuscript.

Referee C:

I am satisfied with the changes and recommend to publish the paper as it is.

We thank the reviewer for the very helpful comments which allowed us to improve our manuscript.